# Osteoarthritis Changes Hip Geometry and Biomechanics Regardless of Bone Mineral Density—A Quantitative Computed Tomography Study

**DOI:** 10.3390/jcm8050669

**Published:** 2019-05-12

**Authors:** Jerzy Narloch, Wojciech M. Glinkowski

**Affiliations:** 1Chair and Department of Orthopaedics and Traumatology of the Locomotor System, Medical University of Warsaw, 02-005 Warsaw, Poland; jerzy.narloch@gmail.com; 2Polish Telemedicine and eHealth Society, 03-728 Warsaw, Poland; 3Centre of Excellence “TeleOrto” for Telediagnostics and Treatment of Disorders and Injuries of the Locomotor System, Department of Medical Informatics and Telemedicine, Medical University of Warsaw, 02-005 Warsaw, Poland; 4Department of Orthopaedics and Traumatology of the Locomotor System, Baby Jesus Clinical Hospital, 02-005 Warsaw, Poland

**Keywords:** quantitative computed tomography, hip, osteoarthritis, osteoporosis

## Abstract

We aimed to compare proximal femur geometry and biomechanics in postmenopausal women with osteoarthritis (OA) and/or osteoporosis (OP), using quantitative computed tomography (QCT). A retrospective analysis of QCT scans of the proximal femur of 175 postmenopausal women was performed. Morphometric and densitometric data of the proximal femur were used to evaluate its biomechanics. We found, 21 had a normal bone mineral density (BMD), 72 had osteopenia, and 81 were diagnosed with OP. Radiographic findings of hip OA were seen in 43.8%, 52.8%, and 39.5% of the normal BMD, osteopenic, and OP groups, respectively (*p* < 0.05). OA was significantly correlated with total hip volume (r = 0.21), intertrochanteric cortical volume (r = 0.25), and trochanteric trabecular volume (r = 0.20). In each densitometric group, significant differences in hip geometry and BMD were found between the OA and non-OA subgroups. Hip OA and OP often coexist. In postmenopausal women, these diseases coexist in 40% of cases. Both OA and OP affect hip geometry and biomechanics. OA does so regardless of densitometric status. Changes are mostly reflected in the cortical bone. OA leads to significant changes in buckling ratio (BR) in both OP and non-OP women.

## 1. Introduction

The prevalence of osteoarthritis (OA) increases with age [1,2,3,4], as does osteoporosis (OP) [5,6,7]. The relationship between the diseases remain unclear, with variable reports on their coexistence [8,9,10,11].

In adults, bone shape continues to be affected by periosteal apposition (modeling) and endosteal resorption and formation (remodeling), resulting in substantial alteration of bone shape and size. Pathological changes, i.e., OA and OP might add to the dynamics of these [12].

Bone quality could not be solely attributed to BMD (bone mineral density) [13,14,15]. Bone morphology and geometry considerably add to the strength model. Separate assessment of the cortical and trabecular bones is necessary to distinguish the differences in their age-related changes, biomechanics, and response to pharmacological and non-pharmacological treatments. The trabecular bone is about eight times more metabolically active than the cortical bone and is subjected to early and rapid changes with advancing age [13,14].

Quantitative computed tomography (QCT) allows analysis of all bone compartments, facilitating the understanding of the three-dimensional bone structure and strength [12,16,17,18,19,20]. For these reasons, increasing interest in QCT is noted in the literature recently [8,12,13,14,18,19,20,21,22,23,24,25,26].

Both OA [27] and OP [28,29] affect hip geometry and strength, yet no quantitative radiological data are comparing these in the literature. This study was aimed to compare proximal femur geometry and biomechanics between postmenopausal women diagnosed with OA and and/or with OP, using QCT, and to evaluate the extent to which the two diseases coexist in this group of patients.

## 2. Materials and Methods

QCT- scans of the proximal femur of 175 consecutive postmenopausal women presenting with low back (LBP) and groin pain to the emergency department or outpatient clinic were collected.

A multi-detector-row CT (computed tomography) scanner (Aquilion16, Toshiba Medical Systems Corporation, Tokyo, Japan) at the Radiology Department was used for evaluation of both the lumbar spine and proximal femurs. Patients were scanned with conditions adjusted to 120 kV, 250 mA, reconstruction thickness of 0.5 mm, and spatial resolution of 0.625 × 0.625 mm. For the proximal femur analysis, patients were placed on a supine position with the solid calibration phantom (Mindways, Austin, TX, USA) placed beneath the patient between the hips. The region scanned extended just above the femoral head to 3.5 cm below the lesser trochanter. The CT scanner table height was set at the level of the greater trochanter.

Participants were categorized into subgroups due to OA (radiographic signs) and due to OP based on BMD T-scores/or presence of osteoporotic vertebral compression fracture. Patients with chronic endocrine diseases, taking antiresorptive drugs, and after total hip arthroplasties were excluded from the study. The institutional review board approved the study. Informed consent was obtained from the patients before study participation.

Proximal femur analysis was performed using the QCT Pro Bone Investigational Toolkit (BIT, Mindways, Austin, TX, USA) [30]. All volumetric bone mineral density (vBMD) measurements and structural characteristics were extracted from our QCT Pro BIT database. The software separated the cortical bone based on a fixed threshold of 350 mg/cm^3^ for all CT scans. The narrow femoral neck (FN) region was found automatically as a perpendicular plane to the FN axis where the approximate diameter ratio (superior-inferior and anterior-posterior) was 1.4, producing the lowest cross-sectional area (CSA) of the FN. The produced cross-section was then divided automatically into 16 sectors (defined by equal angles of 22.5°) with the origin at the center of the mass. For all sectors, vBMD was assessed separately for trabecular and cortical compartments. FN angle, width, overall CSA, volume, and mass, hip axis length, cross-sectional moment of inertia (CSMI), section modulus (Z), and buckling ratio (BR) were measured using BIT.

The Z is a measure for withstanding bending stress. QCT Pro evaluates section modulus along the strongest (Zmax—from the geometric center to periosteal surface) and weakest axis (Zmin—corresponding periosteal distance along orthogonal to Zmax axis), which combined reflect the ability to withstand torsion. CSMI is a derivative of section modulus, and measures the mass distribution relative to the geometric center reflecting how effective a cross-section is at resisting bending and torsion—depending on the axis chosen for calculation. Both Z and CSMI assume homogenous distribution of cortical bone, however differences in porosity and mineralization lead to varied voxel density. To address this, each voxel’s area is multiplied by the ratio of measured cortical density to physiologic bone density to produce density-weighted Z and CSMI (DW-Z, DW-CSMI).

The BR reflects strength against compressive stress leading to sudden sideways deflection of the structural member. BR is a measure of cortical instability consequential of excessive cortical thinning. BR relates the cortical thickness to the width of the femoral neck.

Apart from the structural characteristics, we also evaluated density measures of the proximal femurs, taking note of any signs of hip joint OA. The OA and sub-OA subgroups were defined based on the hip joint CT assessment of cartilage destruction, and the presence of osteophytes. Lumbar spine scans were evaluated to identify vertebral compression fractures. Diagnosis of OP was based on the QCT BMD criteria of the T score values using the National Health and Nutrition Examination Survey (NHANES) DXA for hip QCT [19].

All of the analyses were carried out using TIBCO Software Inc. (2017) Statistica (data analysis software system), version 13.1. Descriptive statistics of all variables were calculated. Normally distributed quantitative variables were compared using the Student t-test; and non-normally distributed categorical variables were compared using the U-Mann Whitney test and Kruskall–Wallis test. Correlations between variables were calculated using Spearman’s Rank Correlation Coefficient. The statistical level of significance was set at *p* = 0.05.

## 3. Results

### 3.1. Participants’ Baseline Characteristics

In the 175 postmenopausal women included in this study, the mean age was 68.8 years (standard deviation (SD) 11.26 years, standard error (SE) 0.85 years), mean weight was 64.4 kg (SD 14.9 kg, SE 4.87 kg), and mean height was 159 cm (SD 6 cm, SE 0.45 cm). Among the women, 12% had a normal BMD, 41.1% met the densitometric criteria of osteopenia, and 46.9% were diagnosed with OP. Data on patients’ baseline characteristics are summarized in Appendix A.

Radiographic signs of hip OA were seen in 43.8%, 52.8%, and 39.5% in the normal BMD, osteopenic, and OP groups (*p* < 0.05).

Nearly one-half (79 women −45%) of women had radiographic signs of at least unilateral hip OA, whereas 45 (25.7%) sustained at least one vertebral compression fracture in the lumbar spine, with 20 women (25.3%) having OA as a coexisting disease. Overall, 70 (40%) women had decreased BMD (either osteopenia or OP) combined with radiographic hip OA, with 32 women (18.3% overall, 45.7% of the OA group) being osteoporotic.

Spearman’s rank correlation coefficient showed a weak, yet significant association between hip OA and history of hip fracture (not site-matched), and the moderate association between densitometric status and history of vertebral fractures (r = 0.16, *p* < 0.05 and r = 0.28, *p* < 0.05, respectively).

### 3.2. Morphological and Densitometric Findings

A number of morphological and densitometric measures proved to be significantly different between the OA and non-OA subgroups (Appendix A). FN characteristics (angle, width, and height) were not different between the OA and non-OA subgroups, regardless of densitometric status. However, the FN volume was significantly different in the patients with a decrease in overall BMD.

Patients’ physical characteristics were mostly correlated with hip axis length and total hip volume (r = 0.36 and r = 0.46, respectively—*p* < 0.05), mostly reflected in the volume of the intertrochanteric region (r = 0.43, *p* < 0.05). Weight was overall less correlated with hip characteristics but was mostly correlated with cortical indices, such as total hip and intertrochanteric cortical volumes (r = 0.36 and r = 0.38, respectively; *p* < 0.05). Hip BMD measurements did not reach those levels of association.

FN morphology expressed in angle, width, and length was mostly correlated with total hip and trochanteric cortical volumes (r = 0.55 and r = 0.57, respectively; *p* < 0.05). Morphological indices of FN were mostly insignificantly correlated with BMD measurements. Those that reached a level of statistical significance had weak correlations.

### 3.3. Biomechanical Findings

Collective biomechanical data of patients’ subgroups (OA vs. non-OA) are presented in Figure 1. The gradual increase in BR with decreasing BMD can be noted from our data. There was a decrease in Z, CSMI, and CSA across the groups (separately cortical and whole bone).

In the normal BMD group, all indices, except cortical CSMI, were significantly different between the OA and non-OA subgroups. In the osteopenia and OP groups, the differences reach statistical significance in all measurements (Figure 2).

### 3.4. Sectors

Each sector was characterized by its perimeter, average cortical BMD, average trabecular BMD, average cortical thickness, normalized to BMD cortical thickness, the average distance from the center of mass to the cortex, and average distance from the geometric center to the cortex (Figure 3).

In the supero-posterior region (13th–16th sectors), there were no significant differences between the normal BMD and osteopenic groups in the average trabecular BMD. In the posterior overlapping region (10th–14th sectors), there were no significant differences between the osteopenic and OP groups in the average cortical BMD. All other measurements of the abovementioned characteristics were statistically significant among the normal BMD, osteopenic, and OP groups.

Spearman’s rank correlation coefficient showed no significant association between age and cortical/trabecular BMD in the 10th–12th sectors of the femoral neck (infero-posterior part). At the same time, significant correlations with average cortical thickness were found across all sectors. The highest correlations were produced by adjacent sectors of the super-posterior/superior region [i.e., 15th, 16th, 1st, and 2nd sectors (r = −0.20 to r = −0.22, *p* < 0.05)]. Intersector correlations were the strongest between adjacent sectors (of the same region), in terms of average cortical thickness (up to r = 0.84–0.85 for the 8th, 9th, and 10th sectors), average trabecular BMD (up to r = 0.7–0.76 for the 4th, 5th, and 6th sectors), and average cortical BMD (up to r = 0.85 for the 15th and 16th sectors). Among the individual sectors, the average trabecular BMD of the 10th sector was least significantly correlated with measurements of other sectors. Most commonly, inverse intersector correlations between average trabecular and cortical BMD were noted across all sectors.

## 4. Discussion

In this study, we retrospectively investigated the coexistence of OP and hip OA in postmenopausal women, and their quantitative effect on proximal femur geometry and biomechanics. Associations between proximal femur three-dimensional architecture, cortical bone geometry and strength were presented in previous studies [15,31]. Yet, there are no reports comparing OA and OP.

In this studied cohort, we showed the prevalence of OP in the group with radiographically proven hip OA (45.7%); of these, >25% sustained a vertebral compression fracture. If we include osteopenia, the conditions coexist in 70 of 79 women with OA, which is higher compared to that reported in large prospective population-based cohorts (20.7–28%).

Our research focused on postmenopausal Caucasian women. Borggrefe et al. [12] investigated a large cohort of older men, which were categorized according to the history of hip fracture. QCT-derived measures of the femoral neck region showed more correlation between vBMD and Z or BR (Z − r = 0.47 vs. r_normal_ = 0.13, r_osteopenia_ = 0.39, r_osteoporosis_ = 0.64; r_normal_ was not statistically significant; BR − r = −0.79 vs. r_normal_ = −0.81, r_osteopenia_ = −0.72, r_osteoporosis_ = −0.64). Both groups were older than the women in our study, regardless of the densitometric status (mean 73.3–77.1 years vs. 63.1–71.8 years). This discrepancy implies that gender-dependent proximal femur geometry contributes significantly to the ability to withstand stress. Indeed, Yates et al. found in their hip structural analysis (HSA)-based study, significant gender differences in hip structural geometry [32].

Furthermore, the differences increased with age. The differences were seen in CSA (cross-sectional area), outer diameter, cortical thickness, Z, and BR in both the femoral neck and intertrochanteric regions. The findings were subsequently confirmed in a large QCT-based prospective population study [33]. There is a recognized tendency of femoral neck expansion [8,12,32]. Periosteal apposition leads to an increasing CSA of the femoral neck with age. The endosteal expansion resulting in widening of the endosteal cavity can impact the stability of the femoral implant. In our group, cortical CSA of the femoral neck decreased with decreasing BMD. Total hip volume (cortical and trabecular bone volumes combined) differed between non-OP and OP women (266.1 cm^3^ vs. 198.7 cm^3^), but the difference was non-significant. When comparing their hip compartments, both groups showed significant difference. Overall, we observed, contrary to previous reports [8,12,32] that the OP group was characterized by smaller volumes of different hip regions (total hip, femoral neck, greater trochanter, intertrochanteric zone, and Ward’s triangle). This could be partially attributed to the less robust physical characteristics (both weight and height). Indeed, these were mostly correlated with hip axis length and total hip volume and mostly reflected in the volume of the intertrochanteric region [15]. Weight was overall weakly but significantly correlated with hip characteristics, mostly with cortical indices such as total hip and intertrochanteric cortical volumes.

The differences in the presence of OA in each densitometric group showed significantly higher volumes in the cortical compartments in most regions in women with OA. The differences were not conspicuous in trabecular compartments or overall. The presence of osteophytes contributing to the increased in cortical volume might explain these observations. Volumetric data were reflected in the biomechanical measurements for all densitometric groups. The OA affected hips showed better mechanical strength. The angle, width, and height of the femoral neck did not show any differences between OA and non-OA subgroups, regardless of the densitometric status. The femoral neck is an area that usually does not encompass osteophytosis.

Although the superior-posterior region undergoes gradual cortical thinning, the infero-posterior region of the femoral neck cross-section is least likely to be affected by age-related changes. This is in accordance with previous observations of femoral neck fractures, in which the decline in cortical thickness and density of the superior half of the femoral neck averaged to 3.3% per year and 1.2% per year, respectively, which is in contrast to losses of 0.9% per year and 0.4% per year, respectively, in the inferior femoral neck [23,34]. When comparing the normal BMD and osteopenic groups, there were significant differences in the average trabecular BMD in the superior-posterior region, which is a tendency not seen between the osteopenic and OP group. Thus, initially, the bone stock seemed to deplete significantly in the trabecular compartment, whereas cortical thinning was more noticeable with decreasing BMD. This observation was mentioned only once recently by Khoo et al. [8], who reported a quadratic vs. linear loss of volumetric BMD in the cortical and trabecular compartments, respectively.

QCT relative to DXA (dual-energy X-ray absorptiometry) allows analysis of all bone compartments. It is more sensitive in detecting diminished BMD, since the measurement is not affected by obesity, degenerative changes, joint space narrowing, calcifications and osteophytes [35,36,37,38]. It facilitates the understanding of the three-dimensional bone structure, which can be helpful in preoperative planning [39,40,41], and therapy monitoring—either medical or implant-focused.

Diagnosis of OA could not exclude the diagnosis of OP, especially in the elderly. OP in OA patients requires medical attention. Proximal femur or vertebral compression fracture could be the eventual consequence of low bone mineral density, adding to overall morbidity and mortality in these patients. Untreated OP patients undergoing THA (total hip arthroplasty)have higher intra- and post-operative risks, such as those of intraoperative fracture, periprosthetic osteolysis with implant migration, and postoperative periprosthetic fracture [42]. Postoperative antiresorptive medication reduce the risk of revision surgery by almost 60% [42]. A novel local osteo-enhancement procedure could serve as a preventive measure against proximal femur fracture [43].

The present study has several limitations. First, our analysis was based on QCT findings, which have recognized technical limitations particularly concerning partial volume effects in the cortical regions, which are caused by the limited spatial resolution, and overall beam hardening artifacts, which can influence the measurements [44]. Second, the study included Caucasian women; thus, the results cannot be fully extrapolated to other populations. Despite its insufficiencies, the study has its strengths. It was conducted with a cohort of considerable size. Secondly, it raises an issue generally neglected in studies on OA and OP, wherein patients presenting with specific complaints (in our case LBP and groin pain) may prompt the treating physician to consider that the symptoms could be caused by osteoporotic vertebral fractures (a consequence of OP), which requires prompt medical attention, both general and, if OA patients are to referred to joint arthroplasty, implant oriented.

## 5. Conclusions

Hip OA and OP often coexist. In postmenopausal women, these diseases may coexist in 40% of cases. Both OA and OP differently affect hip geometry and biomechanics. OA does so regardless of densitometric status, yet the discrepancy increased with a decline in bone stock. Changes are mostly reflected in the cortical bone—total hip cortical BMD and volume, intertrochanteric cortical BMD and volume, and CSA, especially. Sectoral analysis showed cortical thinning in the superior-posterior region of FN in women with OP, while osteopenia initially leads to trabecular loss in the same region. In terms of biomechanics, OA leads to significantly notable decrease in BR of both OP and non-OP women, and significant increase in Z and CSMI. QCT clearly shows the density and the architecture of the proximal femur from a broader perspective to researchers as well as orthopedic surgeons and practicing clinicians.

## Figures and Tables

**Figure 1 jcm-08-00669-f001:**
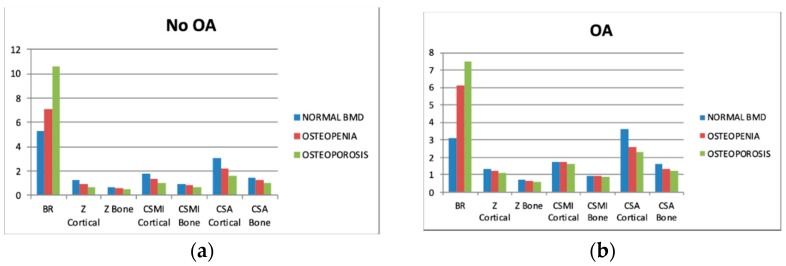
Differences between cross-sectional area (CSA) and biomechanical characteristics, depending on densitometric status—separate graphs for patients with (**1b**) and without (**1a**) osteoarthritis (OA). Please refer to “Methods”. Cross-sectional moment of inertia (CSMI).

**Figure 2 jcm-08-00669-f002:**
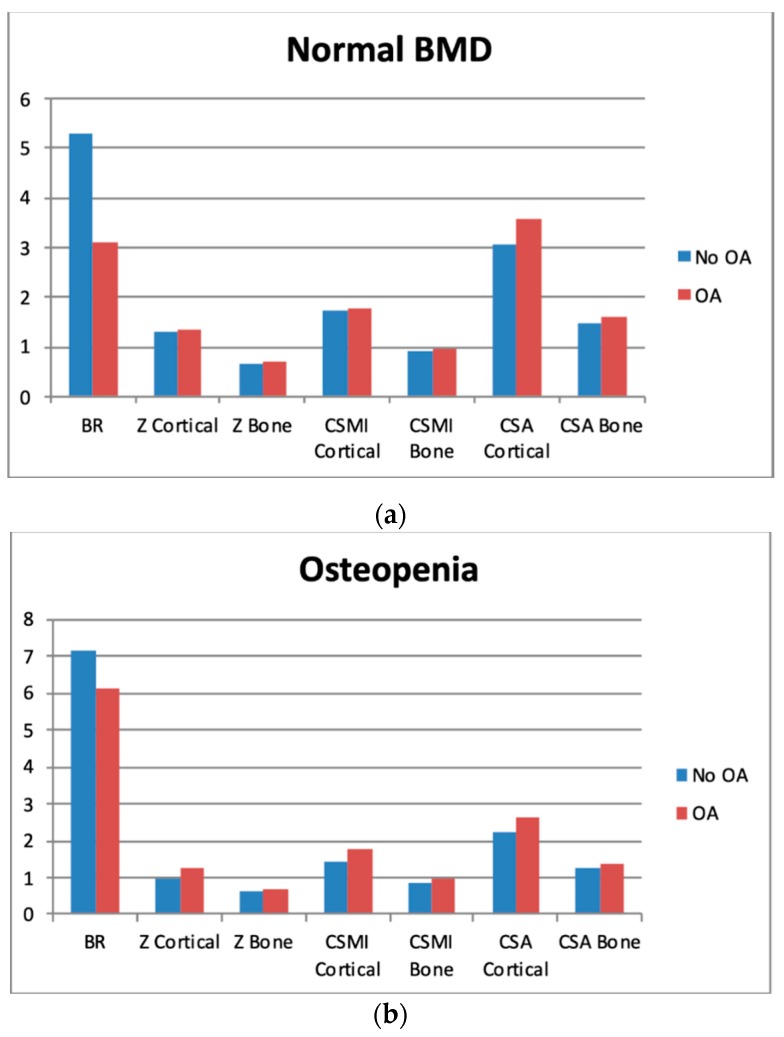
Differences between cross-sectional area (CSA) and biomechanical characteristics, depending on the presence of osteoarthritis (OA)—separate graphs for patients with normal BMD (**2a**), osteopenia (**2b**), and osteoporosis (**2c**). Please refer to “Methods”.

**Figure 3 jcm-08-00669-f003:**
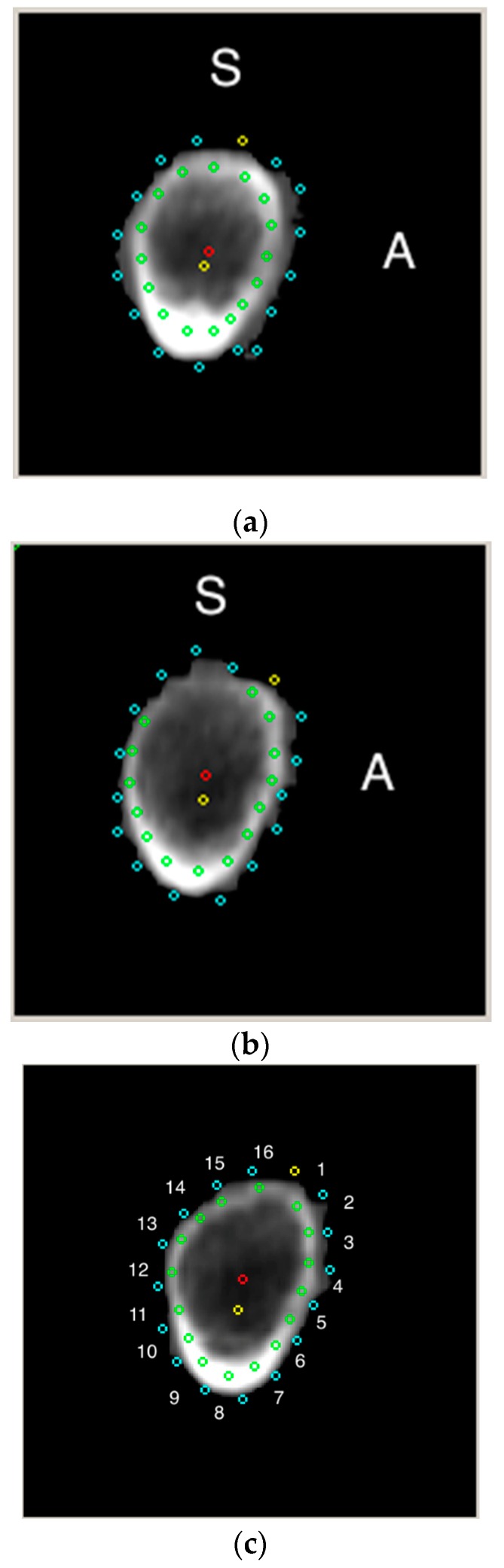
The narrow femoral neck region was found automatically as a perpendicular plane to femoral neck axis where the approximate diameter ratio (superior-inferior and anterior-posterior) was 1.4—producing the lowest cross-sectional area of the femoral neck. Note marked boundaries of 16 sectors (from the most top, clockwise—Figure 3c). Figure 3a - a cross-section of patient with normal BMD, in Figure 3b analogous region in a patient with osteoporosis. Note the diminished cortical thickness in postero-superior region with decreased BMD. S—superior, A—anterior, red dot—geometric center, yellow dot—center of mass.

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
