# Peer review of "Osteoarthritis Changes Hip Geometry and Biomechanics Regardless of Bone Mineral Density—A Quantitative Computed Tomography Study"

_jcm, 2019, doi:10.3390/jcm8050669_

Round 1
Reviewer 1 Report
Comments
In this study the authors looked at the proximal femur geometry in postmenopausal women with OA and/or OP via QCT. The results were of interest and helps a clinician presented with complaints such as LBP and groin pain in these patient populations think of other possibilities. I also admire the honesty of the authors concerning the limitations of the study and the reflection of non- significant data.
Text edits
Please define the abbreviation of BMD in full in line 36.
Please correct the extra gap on line 91 between the full stop and the word Lumber.
Please reformat the graphs to make the legend text and axis text readable. They are too small and come across as blurs. It is hard to read them even when the pdf is zoomed in.
Please state tables 1 and 2 are in the supplementary figures.
Author Response
Dear Reviewer,
Thank you for your letter and for the comments concerning our manuscript: “Osteoarthritis
changes hip geometry and biomechanics regardless of bone mineral density — A quantitative computed tomography study”
The critique was constructive and relevant. The comments provided significant guidance to our study. We have carefully studied your comments and have made the recommended corrections and revisions. We hope that our revised manuscript is acceptable after the revision.
Please find below our corrections. The responses to your comments are as
follows:
1. Please define the abbreviation of BMD in full in line 36.
Our response: Necessary changes were made.
2. Please correct the extra gap on line 91 between the full stop and the word Lumber.
Our response: Necessary changes were made.
3. Please reformat the graphs to make the legend text and axis text readable. They are too small and come across as blurs. It is hard to read them even when the pdf is zoomed in.
Our response: Figures were reformatted.
4. Please state tables 1 and 2 are in the supplementary figures.
Our response: Necessary statement was added.
Kind regards,
Authors

Reviewer 2 Report
Dearest Authors,
your manuscript is surely interesting as it addresses a topic of growing interest in orthopedic clinical medicine, also in relation to growing population age trends.
I would suggest minor revisions prior to accepting it for pubblication, particularly, a more detailed biostatistical analysis should be performed on this cohort of patients: absence of biases and a verification that it is a reliable and indicative subset of the target population should be verified, alsowith respect to literature references.
a couple of more framework reviews should be referred in the intoduction and conclusions should be widened.
few typos need to be corrected.
all the best
Author Response
Dear Reviewer,
Thank you for your letter and for the comments concerning our manuscript: “Osteoarthritis
changes hip geometry and biomechanics regardless of bone mineral density — A quantitative computed tomography study”
The critique was very constructive and relevant. The comments provided significant guidance to our study. We have carefully studied your comments and have made the recommended corrections and revisions. We hope that our revised manuscript is acceptable after the revision.
Please find below our corrections. The responses to the reviewer’s comments are as
follows:
1) “I would suggest minor revisions prior to accepting it for publication, particularly, a more detailed biostatistical analysis should be performed on this cohort of patients: absence of biases and a verification that it is a reliable and indicative subset of the target population should be verified, also with respect to literature references.”
Our response: We calculated the standard errors of descriptive data to prove it is indicative of target population. We have compared our population with those described in the literature in the discussion (ref.: Borggrefe, J et al. Osteoporotic Fractures in Men Study Research, G., Association of 3D Geometric Measures Derived From Quantitative Computed Tomography With Hip Fracture Risk in Older Men. JBMR 2016, 31 (8), 1550-8. and Yates, L. B.et al, Hip structural geometry in old and old-old age: similarities and differences between men and women. Bone 2007, 41 (4), 722-32.)
2) “a couple of more framework reviews should be referred in the introduction and conclusions should be widened.”
Our response: We added 3 additional references in the introduction and expanded conclusions.
3) “few typos need to be corrected”
Our response: Necessary changes were made.
Kind regards,
Authors
